# MMTryon: Multi-Modal Multi-Reference Control for High-Quality Fashion Generation

## Abstract

This paper introduces MMTryon, a multi-modal multi-reference VIrtual Try-ON (VITON) framework, which can generate high-quality compositional try-on results by taking a text instruction and multiple garment images as inputs. Our MMTryon addresses three problems overlooked in prior literature: 1) **Support of multiple try-on items.** Existing methods are commonly designed for single-item try-on tasks (e.g., upper/lower garments, dresses). 2) **Specification of dressing style**. Existing methods are unable to customize dressing styles based on instructions (e.g., zipped/unzipped, tuck-in/tuck-out, etc.) 3) **Segmentation Dependency**. They further heavily rely on category-specific segmentation models to identify the replacement regions, with segmentation errors directly leading to significant artifacts in the try-on results. To address the first two issues, our MMTryon introduces a novel multi-modality and multi-reference attention mechanism to combine the garment information from reference images and dressing-style information from text instructions. Besides, to remove the segmentation dependency, MMTryon uses a parsing-free garment encoder and leverages a novel scalable data generation pipeline to convert existing VITON datasets to a form that allows MMTryon to be trained without requiring any explicit segmentation. Extensive experiments on high-resolution benchmarks and in-the-wild test sets demonstrate MMTryon's superiority over existing SOTA methods both qualitatively and quantitatively. MMTryon's impressive performance on multi-item and style-controllable virtual try-on scenarios and its ability to try on any outfit in a large variety of scenarios from any source image, opens up a new avenue for future investigation in the fashion community.

## 1 Introduction

VIrtual Try-ON (VITON) technology aims to generate photo-realistic images featuring individuals adorned in specific garments according to input images of both the garments and the person. Due to its promising potential to revolutionize the online shopping experience, VITON has garnered considerable attention both in the academic and industry communities. Despite the substantial progress achieved in academic benchmarks (Choi et al., 2021; Davide et al., 2022), previous methods (Xie et al., 2021b; Ge et al., 2021; He et al., 2022; Bai et al., 2022; Xie et al., 2023; Morelli et al., 2023; Gou et al., 2023) still fall short of meeting the requirements of real-world applications, in particular for: 1) **Multiple-garment Compositional Try-On**: Users should be able to freely choose and combine a diverse range of tops, bottoms, and accessories from in-the-wild images, rather than being confined to the single-garment try-on scenario. 2) **Flexible Try-On Styles**: Users should have the flexibility to select their preferred try-on style, allowing them to specify how a garment is worn, such as deciding whether a top should be tucked into the trousers or draped over them.

Due to their focus on single garments, these methods encounter challenges when multiple garments and accessories need to be changed simultaneously. This limitation arises because collecting training data for single-garment try-on is relatively easy, whereas gathering data for multi-garment try-on is significantly more difficult. Additionally, these methods struggle to model the subtle differences in how each piece of clothing is styled during a compositional try-on, such as the overlapping relationship between tops and bottoms. Furthermore, given a flat in-store image of a coat with a zipper, existing methods are limited to replicating the garment in its original state onto the target, without the ability to adjust for open or closed styles.

Figure 1: MMTryon can follow complex instructions to generate High-quality try-on results.

Additionally, existing methods lack the capability to precisely identify try-on areas, often resorting to trained segmentation models (Gong et al., 2019; Li et al., 2020) for explicit try-on area localization. This reliance on segmentation models makes these approaches highly dependent on the segmentation model's precision, which can be unsatisfactory in complex scenarios, introducing additional noise during training. Additionally, conditioning the generation on the segmented garment results in a loss of information of how the garment is worn, leading to highly unrealistic try-on outcomes, particularly for uniquely designed garments. Moreover, the dependence on segmentation models severely restricts the flexibility of the try-on process, as these models are typically confined to a predefined set of garment classes.

To address these challenges, we propose MMTryon, a more flexible try-on model capable of achieving multi-modal compositional dressing without the need for pre-segmenting the try-on areas. In our approach, a general try-on scenario is decoupled into text and image information, where the text information encompasses high-level garment descriptions and try-on style instructions, while the image information contains fine-grained texture details. MMTryon is based on a conditional diffusion model that, to ensure efficient incorporation of both the image and text information in challenging scenarios, leverages our novel multi-reference image attention and multi-modal instruction attention modules. Specifically, to exploit the image information, we pre-trained a robust garment encoder on a large dataset to project image features into the multi-reference image attention module. This module then can utilize text as queries to extract the necessary features with the aim of eliminating our dependency on segmentation. For the text information, we develop a sophisticated multi-modal prompt and employ a pre-trained text encoder to project the text information into the specifically designed Multi-Modal Instruction Attention module. Finally, we train our model on an synthetically enhanced dataset, facilitated by our proposed data pipeline, using the pre-trained encoders and Stable Diffusion-initialized weights. Despite the difficulty of obtaining ideal multi-garment-to-single-model pairs for compositional try-on, our pre-trained garment encoder ensures excellent results even with limited datasets. Additionally, by overcoming the segmentation accuracy limitations in existing VITON models, our model can more effectively handle the nuances of different clothing items and styles (see Fig. 1).

Overall, our contributions can be summarized as follows: First, we propose MMTryon which, to our knowledge, is the first model to support the multi-modal compositional try-on task. MMTryon enables the combination of one or multiple garments for try-on, while also allowing the manipulation of try-on effects through textual commands. We further introduce a scalable data generation pipeline and a specially designed garment encoder. This eliminates the need for any prior segmentation networks during both the training and inference phases to get garments, relying solely on text, images of individual garment items, or model images to identify the areas of interest. Finally, extensive experiments conducted on public try-on benchmarks and in-the-wild test sets are performed, demonstrating that our approach surpasses existing state-of-the-art methods such as Outfit Anyone(Sun et al., 2024).

## 2 RELATED WORK

**GAN-Based Virtual Try-on.** Traditional virtual try-on methods (Wang et al., 2018; Dong et al., 2019; Yang et al., 2020; Ge et al., 2021; He et al., 2022; Choi et al., 2021; Xie et al., 2021b; Zhao

et al., 2021; Xie et al., 2021a; Lee et al., 2022; Bai et al., 2022; Davide et al., 2022; Xie et al., 2022; Dong et al., 2022; Huang et al., 2022; Xie et al., 2023) usually adopt a two-stage pipeline with Generative Adversarial Networks (GANs). The first stage employs an explicit warping module to deform in-shop clothing to the target shape, while the second stage uses a GAN-based generator to fuse the deformed clothing onto the target person. For theses methods, the synthetic quality largely depends on the quality of the deformation in the first stage, prompting current methods to emphasize enhancing the non-rigid deformation capabilities of the warping module. Meanwhile, some GAN-based approaches (Cui et al., 2021; Xie et al., 2022) have explored the compositional try-on task, utilizing a cyclical generation mode to achieve sequential try-on. However, when the outcome of compositional dressing is entirely determined by the try-on order, the lack of understanding of the semantic information of the garments often leads to unrealistic results. Furthermore, most GAN-based solutions mentioned above directly utilize UNet-based generators for try-on synthesis, with little exploration into how to enhance the generator's capabilities. This results in poor resolution and visual quality of the try-on outcomes, making it difficult to integrate them with diverse linguistic instructions.

**Diffusion-Based Virtual Try-on.** Compared to GAN-based models, diffusion models have made significant strides in high-fidelity conditional image generation (Rombach et al., 2022; Saharia et al., 2022; Ramesh et al., 2022). Image-based virtual try-on is essentially a specialized form of the image editing/repair task conditioned on a given garment image. Therefore, a straightforward adaptation is to extend text-to-image diffusion models to accept images as conditions. While methods such as Yang et al. (2023); Huang et al. (2023); Chen et al. (2023) have demonstrated their capabilities in virtual try-on, they fall short in preserving the texture details of the try-on results due to inadequate extraction of fine-grained features by the image encoder. Recent methods (Morelli et al., 2023; Gou et al., 2023) have further aimed to integrate traditional GAN-based approaches with diffusion models. They employ explicit warping modules to create deformed garments and use diffusion models to blend these with reference person images. Beyond simply incorporating explicit warping modules, the newly introduced TryonDiffusion (Zhu et al., 2023), conversely, implements an implicit warping mechanism for clothing deformation. This addresses the issues of texture misalignment mentioned earlier and achieves promising try-on synthesis. However, TryonDiffusion relies on precise parsing to achieve various conditions, making it unable to handle more complex try-on scenarios. Additionally, none of the existing approaches have effectively leveraged the capabilities of current large multi-modal models and have discarded text as an input, lacking the ability of free-form instruction following. The try-on effect in these approaches is solely inferred from the garment image.

Recent methods have also explored compositional try-on tasks. For example, MMVTON Zhu et al. (2024) utilizes multiple garments as input, but it only supports label-level text input and cannot control free-text attributes such as the background. AnyFit Li et al. (2024) uses multiple flat-lay garment images as input but lacks additional control over wearing styles. Wear Any Way Chen et al. (2024) incorporates point-based input to achieve style control, while ImageDressing Shen et al. (2024) employs lightweight training methods for embedding injection, enabling flexible results. While these methods have made progress in exploring related tasks, they often rely heavily on prior parsing networks, which may lead to unintended outcomes due to the dependency on precise parsing.

Our approach instead, through the specialized design of a parsing free garment encoder and the multi-modal multi-reference attention mechanism, achieves a more versatile try-on experience. This not only addresses the limitations seen in both GAN and diffusion model-based methods but also utilizes the strengths of large multi-modal models, allowing for more detailed and flexible adaptation to various clothing styles and try-on scenarios.

## 3 METHOD

As shown in Fig. 2, MMTryon has a minimal input setup during inference, only requiring a source person image, multiple reference images, and a text instruction describing how the items from the references are fitted onto the desired person. To achieve multi-modal multi-reference controllable try-on, two key elements are central: the training data and the model design. We start by presenting the design of our model to support this inference setup before describing how the data is gathered to support model training.

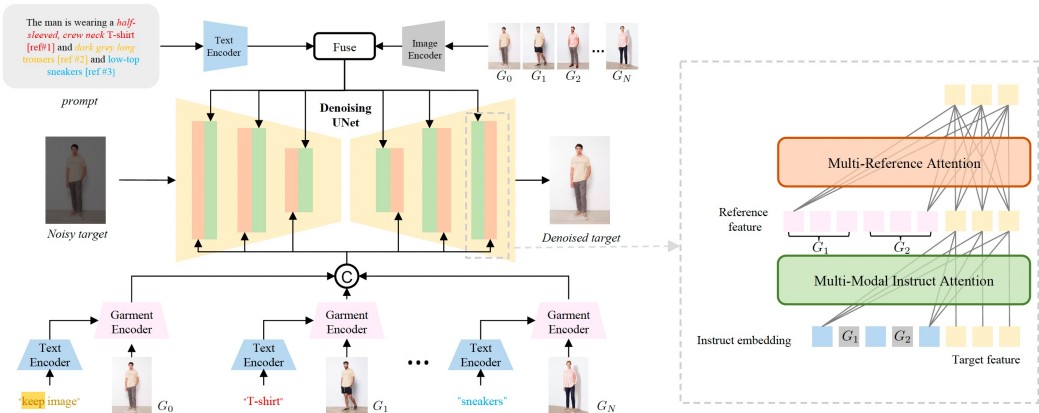

Figure 2: Overview of the proposed MMTryon framework. The instruction prompt and garment images are combined to obtain a multi-modal instruct embedding, replacing the original textual condition. Each garment image together with the corresponding text span are further processed by the garment encoder to obtain reference features, which, along with target features, undergo multi-reference attention to ensure detailed texture transfer.

## 3.1 OVERVIEW

Given a target person image $I$, multiple reference images containing try-on items $\{G_i\}$ and a text instruction $T$ describing how the items from those reference images are dressed, MMTryon aims to seamlessly follow $T$ to synthesize photo-realistic try-on results $I'$. We model this task as a conditional diffusion model, which takes $I$, $\{G_i\}$, and $T$ as conditions and gradually estimates a denoised version of $I'$ from pure noise. Our diffusion model is built upon the public available Stable Diffusion text-to-image model (Rombach et al., 2022).

Stable Diffusion is a text conditioned latent diffusion model (Rombach et al., 2022). For an VAE (Kingma & Welling, 2013) encoded image latent feature $z_0$, the forward diffusion process is performed by adding noise according to a predefined noise scheduler $\alpha_t$(Ho et al., 2020):

$$q(z_t|z_0) = \mathcal{N}(z_t; \sqrt{\alpha_t}z_0, (1 - \alpha_t)I). \tag{1}$$

To reverse the diffusion process, a noise estimator $\epsilon_\theta(\cdot)$ parameterized by an UNet is learned to predict the forward added noise $\epsilon$ with the objective function,

$$\mathcal{L}_{\text{dm}} = \mathbb{E}_{(z_0,c)\sim D}\mathbb{E}_{\epsilon\sim\mathcal{N}(0,1),t} \left[||\epsilon - \epsilon_\theta(z_t, t, c)||^2\right], \tag{2}$$

where $c$ is the text condition associated with image latent $z$, and $D$ is the training set.

In MMTryon, our condition is not merely a text condition but a multimodal condition that includes texture information as well as a multimodal prompt that integrates global image information and instructions. Modulating this condition into the diffusion model is a challenging problem. Prior works (Yang et al., 2023; Chen et al., 2023) have relied on transforming pixel features into text-aligned features using a CLIP encoder and injecting them into the cross-attention layers. However, this approach often results in a loss of attention to detail. In this work, we instead draw inspiration from Hertz et al. (2022); Ge et al. (2023); Cao et al. (2023), who show that the cross-attention layer learns correspondences between the image features and the text embedding tokens, generating an overall image semantic structure, while the self-attention layers are more related to detailed texture generation. We thus focus on enhancing the cross-attention layers conditioned on the original text, integrating it with our multi-modal instruction to improve generation of the final try-on results. Additionally we train a garment encoder to align pixel features with the stable diffusion features, enhancing the self-attention layers and seamlessly transfer detailed image features from the garment images to the desired human images.

## 3.2 PRETRAIN GARMENT ENCODER

As mentioned before, one of the main challenges of multi-reference try-on is the lack of training data, i.e., most of the garment person pairs consist of only a single garment. We draw inspiration

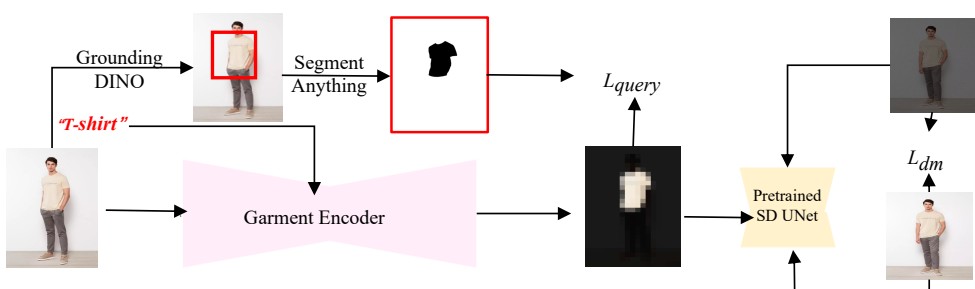

Figure 3: Overview of the proposed pretrained garment encoder.Our garment encoder utilizes a prior mask derived from grouding dino and SAM to improve text query accuracy through cross-attention between the target text and the input features. The garment encoder is supervised by the diffusion reconstruction loss and our text query loss.

from the training scheme of Large Multi-modal Models (e.g., Liu et al. (2024)), and split the training into pretraining and supervised fine-tuning stages. During pretraining, our goal is to align features from multi-sources to the generator, here, the StableDiffusion denoising UNet. Thus we fixed the denoising UNet, and pretrain the garment encoder in this stage.

To achieve texture-consistent try-on results without relying on a prior segmentation model, this encoder should extract fine-grained image features and respond effectively to text queries. We initialized the garment encoder with the UNet encoder from the diffusion model, which takes both garment image and text query as input, we use the features after the cross-attention layer of each transformer blocks as the output garment feature. This feature is then injected into the diffusion UNet through multi-reference attention. By supervising with diffusion loss $\mathcal{L}_{\text{dm}}$, we are able to obtain features that can preserve texture consistency.

Meanwhile, to improve the accuracy of text queries, we enforce the text-queried features to have a similar response to a prior segmentation model SAM (Kirillov et al., 2023) (see Fig. 3). Here we also introduce a text query loss, where features outside the region of the mask are passed through a sigmoid, squared, and averaged to encourage minimization of activations in areas unrelated to the text. The final loss is:

$$\mathcal{L}_{\text{enc}} = \mathcal{L}_{\text{dm}} + \mathcal{L}_{\text{query}} \tag{3}$$

where the text query loss $\mathcal{L}_{\text{query}}$ is:

$$\mathcal{L}_{\text{query}} = \frac{1}{N} \sum \sigma(F \odot (1 - M))^2 \tag{4}$$

Here, $\odot$ denotes element-wise multiplication and $\sigma$ denotes the sigmoid function. $F$ represents the garment feature and $M$ denotes the prior mask corresponding to the target region's text. $N$ denotes the number of elements in $F$ and $M$. This way, we can obtain a garment encoder that can extract only the necessary features corresponding to the text query which also preserves low-level pixel consistency through a combination of discriminative and generative supervision.

### 3.3 MMTRYON

Here, we provide a detailed description of each component within the main UNet. In particular, we employ two types of attention mechanisms to enable multi-modal multi-reference try-on. Having established the powerful garment encoder in the previous section, we now eliminate the dependency on prior masks by utilizing a synthetic dataset to finetune the entire model and the garment encoder jointly. This dataset is derived from the original data through an enhanced data generation pipeline, which is detailed in Sec. 3.3.1.

#### 3.3.1 SCALABLE DATA GENERATION PIPELINE

To achieve multi-modal multi-reference try-on, a straightforward way is to construct training data in the same format. However, existing try-on datasets in this format are not readily available. There are generally two types of datasets, in-shop garment to person pairs or person to person pairs, where

Figure 4: The data generation pipeline of MMTryon. We use a large multi-modal model to describe the target person image, followed by open-vocabulary grounding and segmentation models to extract correspondences between a person image and several garment subjects. For each subject, we utilize SDXL inpainting to obtain the enhanced dataset, which serves as our training data.

the former one only contains a single reference and is usually limited in garment types. Leveraging the person-to-person pairs instead, a possible way to automatically construct a training dataset for the multi-modal multi-reference try-on task is to use segmentation models to crop various reference garments to construct multi-reference pairs. However, the drawback of this approach is that the style of the garments is not controllable and that the model will still be dependent on off-the-shelf segmentation models during inference.

In this work, we instead propose to leverage the increasingly powerful capabilities of large models to develop a flexible scheme for dataset construction. Starting from existing person-to-person datasets, given a pair of person images $I_a, I_b$, we use SOTA large multi-modal model GPT-4V (Achiam et al., 2023) and CogVLM (Wang et al., 2023) to caption the image in a desired format describing how the person is dressed. We then extract the main garment subjects and style (garment categories) from the image description denoted as $\langle obj_1, obj_2, \ldots, obj_n \rangle$. Subsequently, we use a SOTA open-vocabulary detection model (Liu et al., 2023) to generate the bounding boxes corresponding to each garment subjects within the image. The SAM model (Kirillov et al., 2023) is then used to generate precise mask annotations. After this caption-grounding-segmentation procedures, we now established a correspondence between a target person image $I_a$ with description, and all its segmented garments from $I_b$. We find that this process performs well for the vast majority of clothing categories. Note that we need to establish correspondences between two different poses, otherwise this try-on task will degenerate to a copy-paste problem.

To enable parsing-free inference, starting from those garment images, we then use the inpainting ability of the Stable Diffusion model (we use SDXL (Podell et al., 2023) here) with a given prompt to randomly generate an image where a person is wearing such a garment. We use random seeds and random prompts to generate multiple samples, and discard samples with high similarity to prevent information leakage. This way, by iterating through all garment subjects, we are able to construct training pairs, where the target image is described using text, and the reference garments are from different images. The overall process is illustrated in Fig. 4.

### 3.3.2 MULTI-MODAL INSTRUCTION ATTENTION

In our task, it is not possible to ask the user to describe the desired garments from the reference images precisely using only text, as high-level descriptions such as garment type exhibit ambiguity and can not guarantee dressing fidelity. To mitigate this issue, we form a multi-modal instruction using an prompt template:

$$\text{A person wearing} <\text{garment}><\text{style}>[Ref\#1]..., \text{and} <\text{garment}><\text{style}>[Ref\#N]$$

For example, *A person wearing a top, tucked in, [Ref#1], pants [Ref#2] and shoes [Ref#3]*. Here *[Ref#N]* is a special placeholder token prepared for the corresponding reference image. This text prompt is first encoded using CLIP text encoders into text embedding $\mathbf{T} \in \mathbb{R}^{L \times D}$, where $L$ is the length of the tokens and $D$ is the embedding dimension. The embeddings corresponding to the placeholder tokens are then fused with the CLIP image embedding of the related garment images (see Fig. 2). To balance the embedding length and image feature granularity, and also better align the image feature with the text embedding, we do not only use the [CLS] token as in Yang et al. (2023) , but instead use the latent features from the penultimate layers of the CLIP ViT encoder. These latent features are further downsampled using a perceiver-resampler (Alayrac et al., 2022). This multi-modal embedding then serves as the replacement of the original text embedding of the stable

diffusion model and not only aids in the generation of dressing styles but also enhances the stability of the try-on results, ensuring the correct mapping of the garment.

### 3.3.3 MULTI-REFERENCE TEXTURE ATTENTION

The Multi-Modal Instruction Attention cannot fully preserve the details of the garment due to the use of the CLIP ViT image encoder. To guarantee texture consistency, we therefore first prompt the garment encoder with the textual label. The textual label $P_i$ is derived from the instruction prompt $T$ and used to extract features for the particular subject from image $G_i$. After obtaining the features, in each self-attention layer of the diffusion UNet, we extract a set of garment features corresponding to that layer, denoted as $\{F_{G_1}, F_{G_2}, \ldots, F_{G_n}\}$. We denote the original feature map from the diffusion UNet as $F_{\text{target}}$. To perform the attention operation, the query $Q$ is derived from $F_{target}$, while the $K, V$ pairs are obtained from the concatenated features of $\{F_{\text{target}}, F_{G_1}, F_{G_2}, \ldots, F_{G_n}\}$. This multi-reference warping attention allows us to transfer the features of different reference images onto the target image, thus effectively preserving the texture features of each object without mutual interference. The process is illustrated in Fig. 2.

## 4 EXPERIMENT

### 4.1 DATASETS

We conduct extensive experiments on two public high-resolution Virtual Try-ON (VITON) benchmarks, namely VITON-HD (Choi et al., 2021) and DressCode (Davide et al., 2022), as well as on an additional large-scale proprietary e-commerce dataset. Experiments are conducted under a resolution of $1024 \times 768$. Specifically, VITON-HD comprises 13,679 image pairs of front-view upper-body women and upper-body in-store garment images, which are further divided into 11,647 training pairs and 2,032 testing pairs. DressCode includes 48,392 training pairs and 5,400 testing pairs of front-view full-body person and in-store garment images, consisting of three subsets with different category pairs (i.e., upper, lower, dresses). Our proprietary e-commerce dataset contains 203,239 training pairs and 5,219 testing pairs of front-view full-body person and in-store garment images, and 450,931 training pairs and 9,198 testing pairs of multiple front-view images of the same full body.

### 4.2 IMPLEMENTATION DETAILS

During the data augmentation process, we enhanced the data across the three datasets, running the entire enhancement workflow on 16 A800 GPUs for 5 days. The augmented dataset comprises approximately 2 million images. In our MMTryon implementation, we utilize the SD 1.5 pretrained model (Rombach et al., 2022) to initialize the weights of our main UNet and the weights of our pretrained garment encoder. For the encoding of both text and global image information, we use clip-large-14 (Radford et al., 2021).The garment encoder was pretrained on 8 A100 GPUs for 3 days with a total batch size of 8. MMTryon was trained on 8 A100 GPUs for 1 days with a total batch size of 8.

### 4.3 BASELINES

In our study, we initially compare our approach on traditional try-on tasks with previous baseline methods. We selected four advanced GAN-based methods, namely PF-AFN (Ge et al., 2021), FS-VTON (He et al., 2022), SDAFN (Bai et al., 2022), GP-VTON (Xie et al., 2023) and the latest diffusion-based methods, DCI-VTON (Gou et al., 2023) and StableVTON (Morelli et al., 2023). We directly utilize their released pretrained models for this comparison. Unfortunately, we are not able to conduct an extensive comparison with TryonDiffusion (Zhu et al., 2023) as it is not publicly available at the time of submission. Secondly, for the compositional and multi-modal try-on tasks, we compare our work with paint by example (Yang et al., 2023), Stable diffusion (Rombach et al., 2022), and DALLE3 (Betker et al., 2023). For all baselines, we strictly adhere to the official instructions for running their training and testing scripts. Additionally, to further demonstrate our models zero-shot capabilities, we further perform a comparisons with the state-of-the-art community model, Outfit Anyone.

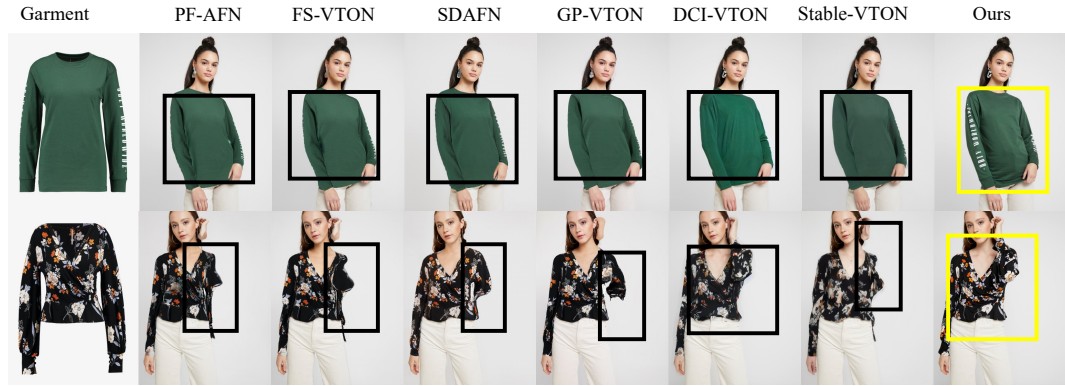

Figure 5: Qualitative comparisons on VITON-HD in the single try-on task. Compared with other methods, our method MMTryon produces more realistic and texture-consistent images.

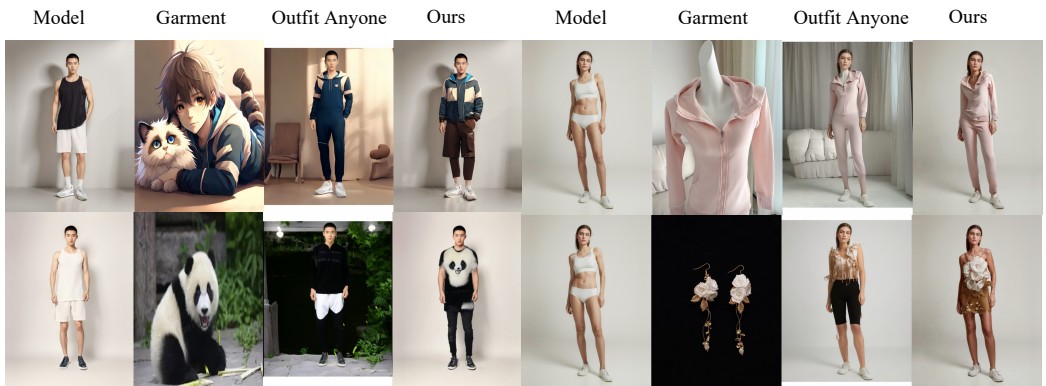

Figure 6: Qualitative comparisons in the wild. Compared with OutfitAnyone(Sun et al., 2024), our method MMTryon produces more realistic and stable results. Note, as Outfit Anyone is only available through their user interface, the comparisons here are limited to their provided model images. Additional result with other baselines are provided in the appendix.

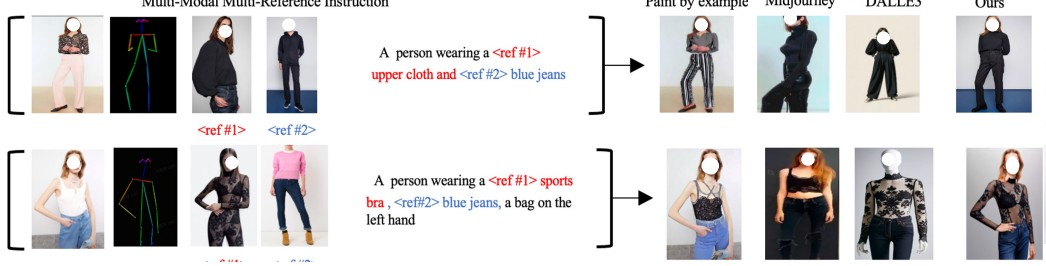

Figure 7: Qualitative comparisons with Paint-by-Example Yang et al. (2023), Midjourney and DALLE3 Betker et al. (2023) in the Multi-Modal Multi-Reference task. Compared with other methods, our method can achieve the best results both in terms of faithfulness and realism.

## 4.4 QUALITATIVE RESULTS

**Comparison on single garment try-on.** Fig. 5 and Fig. 6 provide qualitative comparison between MMTryon and the state-of-the-art baselines in the single garment try-on task on the VITON-HD dataset (Choi et al., 2021). The results demonstrate the superiority of our MMTryon over the baselines. Firstly, our method employs a more powerful garment encoder, which results in better texture consistency compared to the baseline. Secondly, as our approach utilizes more textual information,

Table 1: Quantitative comparisons on the single garment tryon.

| Method | SSIM ↑ | FID ↓ | LPIPS ↓ | KID ↓ | HE ↑ |
|---|---|---|---|---|---|
| PF-AFN Ge et al. (2021) | 0.885 | 9.616 | 0.087 | 3.85 | 0.03 |
| FS-VTON He et al. (2022) | 0.881 | 9.735 | 0.091 | 3.69 | 0.02 |
| SDAFN Bai et al. (2022) | 0.881 | 9.497 | 0.092 | 2.73 | 0.03 |
| GP-VTON Xie et al. (2023) | 0.893 | 9.405 | 0.079 | 0.88 | 0.10 |
| DCI-VTON Gou et al. (2023) | 0.868 | 9.166 | 0.096 | 1.10 | 0.10 |
| StableVTON Kim et al. (2024) | 0.866 | 8.992 | 0.079 | 1.03 | 0.20 |
| Ootdiffusion Xu et al. (2024) | 0.899 | 8.893 | 0.071 | 0.90 | 0.10 |
| Anydoor Chen et al. (2023) | 0.683 | 12.735 | 0.149 | 5.87 | 0.02 |
| AnyFit Li et al. (2024) | 0.893 | 8.60 | 0.075 | 0.55 | |
| Wear-any-way Chen et al. (2024) | 0.877 | **8.155** | 0.078 | 0.78 | |
| **MMTryon** | **0.912** | 8.702 | **0.069** | **0.58** | **0.40** |

Table 2: Quantitative comparisons on the compositional garment try-on task.

| Task | Compositional Try-on | | | Multi-Modal Try-on | | | |
|---|---|---|---|---|---|---|---|
| Method | FID ↓ | KID ↓ | HE ↑ | FID ↓ | KID ↓ | CLIP Score ↑ | HE ↑ |
| Paint-by-Example | 12.429 | 4.89 | 0.06 | 15.231 | 5.21 | 0.12 | 0.04 |
| Midjourney | 9.231 | 3.41 | 0.09 | 10.243 | 3.45 | 0.08 | 0.10 |
| DALL·E3 | 9.441 | 3.19 | 0.15 | 12.241 | 4.12 | 0.13 | 0.08 |
| **MMTryon (ours)** | **8.902** | **0.47** | **0.7** | **8.187** | **0.42** | **0.30** | **0.78** |

it achieves greater stability in try-on styles than the baseline, showcasing the effectiveness of our method.

**Comparison on multi-modal and compositional try-on.** Fig. 7 provides a qualitative comparison between MMTryon and the state-of-the-art baselines for the compositional try-on and multi-modal try-on tasks in the wild. The results demonstrate the superiority of our approach relative to the baselines. Firstly, for compositional try-on, our method exhibits better texture consistency and more realistic try-on effects compared to the baselines. Secondly, for multi-modal try-on, our approach demonstrates more refined performance in terms of texture consistency. At the same time, baselines often overlook the textual descriptions of dressing styles, whereas our method demonstrates stronger text-image consistency, effectively reflecting the intentions expressed in the text. This demonstrates that our approach achieves state-of-the-art results for the compositional and multi-modal try-on tasks. Additional qualitative results of our method are included in Appendix A.4.

## 4.5 QUANTITATIVE RESULTS

**Metrics.** For traditional single-garment try-on tasks, we utilize four widely-used metrics, namely the Structural Similarity Index (SSIM) (Wang et al., 2004), Perceptual Distance (LPIPS) (Zhang et al., 2018), Kernel Inception Distance (Sutherland et al., 2018), and Fréchet Inception Distance (FID) (Parmar et al., 2022), to evaluate the similarity between synthesized and real images. For the compositional try-on task, we similarly employ FID and KID to measure the quality of image generation. In the context of the multi-modal instruction try-on task, we assess the quality of image generation using FID, while the Clip score is utilized to evaluate the consistency between instructions and the final generated images. Additionally, we conducted a Human Evaluation (HE) study, inviting 100 reviewers to assess the synthesis quality and text-image consistency across single-garment try-on scenarios. More details on the Human Evaluation are included in Appendix A.2.

**Evaluation.** As reported in Tab. 1 and Tab. 2, our MMTryon consistently outperforms the baseline methods across all metrics, proving that MMTryon can generate try-on results with better visual quality in single garment try-on, compositional try-on and multi-modal try-on. Furthermore, in our

source    top    bottom    w/o PR    w/o TQL    MMTryon          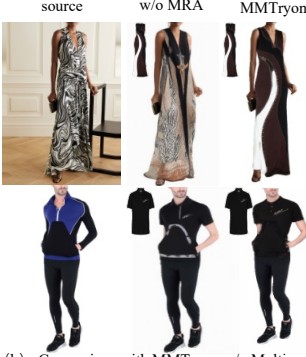

(a) Comparison with MMTryon w/o pretrained garment encoder and MMTryon w/o text query loss          (b) Comparison with MMTryon w/o Multi-Reference Attention

Figure 8: Qualitative results of the ablation study. MMTryon w/o PR indicates MMTryon where only the second stage is trained, MMTryon w/o TQL indicates MMTryon without the text query loss in the first stage, MMTryon w/o MRA indicates MMTryon without the multi-reference attention.

human evaluation, we compare the generated results of the baselines to our model. Higher human evaluation scores indicate that a larger proportion of participants prefer the outcomes of the given method. The human evaluators prefer the results generated by MMTryon in most cases, indicating superior texture consistency and text-image consistency.

## 4.6 ABLATION STUDY

To validate the effectiveness of the pretrained garment encoder, we conduct an ablation study by comparing MMTryon with two ablated versions. One that directly trains the entire model without pretrained garment encoder (denoted as w/o PR) and one that does not include the text query loss (denoted as w/o TQL). The results, as shown in Fig. 3, indicate that omitting the pretrained garment encoder or text query loss will lead to a decrease in garment consistency and overall generation quality. The quantitative results, as presented in Fig. 8, demonstrate that MMTryon with pretrained garment encoder and supervised by text query loss significantly outperforms the other two versions as the garment consistency score and overall generation quality score are both higher. Next, to further verify the effectiveness of the multi-reference attention, we conduct another ablation study. In this study, we compare MMTryon with a version that only utilizes the CLIP features of the clothing, denoted as w/o MRA, where we replaced the multi-reference attention with the original self-attention module. We again compare the main metric scores and observe a decline in both image quality and texture consistency when multi-reference attention is removed. Meanwhile, Fig. 8(b) indicates that rich clothing features are key to maintaining texture consistency. Additionally qualitative ablation results are provided in Appendix A.3.

Table 3: Ablation results demonstrate the benefit of the pretrained garment encoder (PR), text query loss (TQL), and multi-reference attention (MRA).

| Method | FID ↓ | KID ↓ | CLIP Score ↑ |
|---|---|---|---|
| w/o PR | 10.23 | 0.68 | 0.22 |
| w/o TQL | 9.25 | 0.65 | 0.24 |
| w/o MRA | 12.89 | 0.87 | 0.18 |
| **MMTryon** | **8.90** | **0.58** | **0.27** |

## 5 CONCLUSION

In this work, we introduce MMTryon, a novel and powerful try-on model capable of freely generating high-fidelity VITON results with realistic try-on effects based on text and multiple garments. By using a pretrained encoder, MMTryon not only avoids the need for segmentation but also achieves compositional try-on even with limited data. To support multi-modal and multi-reference dressing modes, MMTryon introduces multi-modal instruction attention and multi-reference attention modules. Experiments conducted on high-resolution VITON benchmarks and in-the-wild test sets demonstrate MMTryon's superior efficacy compared to existing methods.

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

# A    APPENDIX

The outline of this appendix is as follows:

- Additional details of the training dataset;
- Details on the Human Evaluation;
- Ablation study;
- Additional visual results of our MMTryon.
- Limitation and future work;
- Source code;
- Societal impact;
- Inference Setting;
- More result for texture and comparision

## A.1    ADDITIONAL TRAINING DATASET DETAILS

The dataset was sourced from authorized e-commerce websites and we have obtained the necessary permissions for the use of the data and the models featured within it. The specific websites are withheld to ensure anonymity, but will be included in the final version.

In the process of creating the dataset, we used seven distinct prompts to guide the SDXL model for inpainting. These prompts mainly involved categories of clothing, such as trousers and skirts, to generate varied garment images. We applied human detection methods to filter out inpainted images of poor quality, retaining only 3-5 of the most realistic and suitable images for the training data.

**Data Visualization and SAM Accuracy**: Figure 9 showcases visualizations of a random subset of the training data. This subset highlights the diversity and quality of the collected inpainted images, while Figure 13(b) illustrates the accuracy of the SAM model in detecting specific garment sections.

**Garment-to-Person and Synthetic Data Generation:** To further enhance the dataset, we trained a single-reference try-on network. Using this model, we transformed the garment-to-person data into person-to-person pairs, generating synthetic garment-person-person combinations. This resulted in a multi-reference multi-modal instruction dataset that combines synthetic multi-reference data with single-reference garment-person pairs.

**Ensuring Dataset Diversity and Quality:** To ensure the dataset is representative of various demographics, we developed a comprehensive labeling system that accounts for attributes such as gender, skin tone, age, and body type. This labeling system aimed to ensure that the model performs well across a wide range of users. Additionally, we utilized **CogVLM** Wang et al. (2023) to annotate and balance the dataset based on various attributes, including garment categories, styles, and other relevant features. Example prompts for this step are provided in Table 4. These garment-person pair pairs represent a wide variety of body types and characteristics.

Table 4: Tag generation questions.

| Question |
| --- |
| Is this image a garment or a model? Use one word. |
| Describe this picture in detail. |
| Describe the picture in detail from the perspective of clothing description, dressing style |
| What is the sleeve length of this garment? Is it long sleeves, short sleeves, or sleeveless? |
| If it's long sleeves, are the sleeves rolled up? |
| How long are the sleeves? Do they reach the wrist, elbow, or shoulder? . |
| Does this garment have a visible logo or brand marking? |
| What is the neckline style? Is it a round neck, V-neck, or another type? |
| Is the overall fit of this garment slim, loose, or some other cut? |
| What is the main color of this garment? Are there multiple colors or accents? |
| Does the garment have a fitted waist, or is it more loose around the waist? Only use words. |
| Are the shoulders narrow, broad, or do they feature any specific design? |

This process resulted in a high-quality dataset consisting of millions of garment-person pairs, which serves as a robust foundation for training and evaluating our model..

Finally, a manual review process was conducted to verify the quality and diversity of the dataset. This process ensured that the dataset met high standards for diversity in lighting, poses, and occlusions, and that no problematic data were included, ensuring robustness across different scenarios.

## A.2 HUMAN EVALUATION DETAILS

For the human evaluation, we designed questionnaires for the single garment try-on, multi-modal garment try-on, and compositional garment try-on. 100 volunteers were invited to complete a questionnaire comprised of 50 assignments for each task. Specifically, for the single garment try-on, given a person image and a garment image, volunteers were asked to select the most realistic and accurate try-on result out of seven options generated by our MMTryon and baseline methods (i.e., PF-AFN (Ge et al., 2021), FS-VTON (He et al., 2022), SDAFN (Bai et al., 2022), GP-VTON (Xie et al., 2023), DCI-VTON (Gou et al., 2023), ,Stable-VTON (Morelli et al., 2023)),anydoor (Chen et al., 2023) and Ootdiffusion (Xu et al., 2024). The order of the generated results in each assignment was randomly shuffled. For the compositional garment try-on, given a person image and a set of garment images, volunteers were asked to select the most realistic and accurate try-on result out of four options generated by our MMTryon and baseline methods (i.e., Paint by example Yang et al. (2023), Midjourney, DALLE3 Betker et al. (2023)). For the multi-modal garment try-on, given a person image, a set of garment images, and a text instruction, volunteers were asked to select the most realistic and instructionally relevant try-on result out of four options generated by our MMTryon and baseline methods (i.e., Paint by example Yang et al. (2023), Midjourney, DALLE3 Betker et al. (2023)).

## A.3 ABLATION STUDY

**Effectiveness of the large-scale Segmentation model:** To validate the necessity of integrating large-scale, open-vocabulary models in instruction-based try-on, we conducted further ablation studies. While domain-specific segmentors may work well in specific cases, relying on them introduces artifacts and errors. To mitigate these issues, we combine the Text-Query Loss and the Reconstruction Loss, allowing MMTryon to train end-to-end. Figure 10(a) illustrates the limitations of traditional segmentors, which are prone to errors from incorrect annotations, highlighting the need for a more robust solution.

**Garment Encoder Without Diffusion Loss**: We performed an additional ablation to evaluate the role of the diffusion loss on the garment encoder's performance. When the diffusion loss was removed,

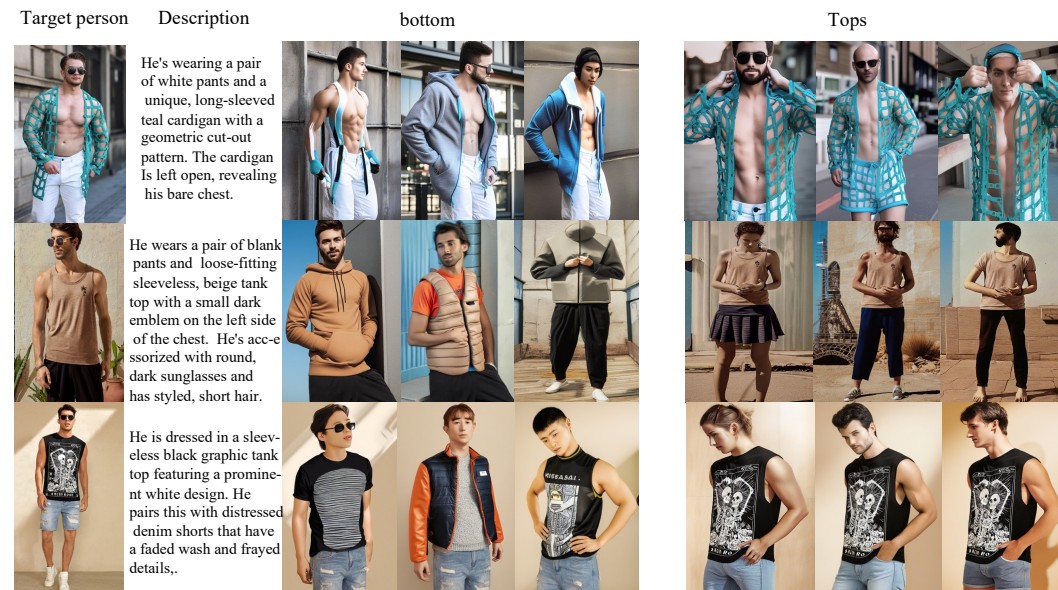

Figure 9: Visualization of our training dataset.

the encoder failed to converge properly. This can be attributed to the lack of alignment with the main UNet, which the diffusion loss helps to maintain. In our experiments, we used only the Text-Query Loss to train the garment encoder. However, as shown in Figure .10(b), the results were suboptimal, underscoring the importance of the diffusion loss in achieving accurate garment encoding.

**Effectiveness of Augmented Dataset:** To further validate the contribution of our augmented dataset, we conducted an ablation study by training a variant of MMTryon (denoted as MMTryon$) using the dataset without high-quality inpainting. Figure 10(c) illustrates that this variant struggled with query accuracy, resulting in less precise try-on outcomes. In contrast, the full MMTryon model, trained with the complete augmented dataset, significantly outperformed its counterpart in understanding and responding to text-based queries. These findings highlight the crucial role of high-quality data augmentation in enhancing try-on performance.

## A.4 ADDITIONAL RESULTS

**Handling variations the in input image**: As shown in Figure 11, our approach effectively handles diverse lighting, poses, and occlusions as the garment encoder, pre-trained on a large dataset, accurately extracts features under various conditions. In the second part of our dataset, repainting prompts were designed to account for lighting and pose diversity. Additionally, by not requiring segmentation inputs in the second stage, our method avoids common errors in occluded scenarios, ensuring robustness and performance without added noise.

**Additional result**

Fig 12 provides additional comparison results. Fig 13(a) show the results for overlapping clothes. Fig 14 demonstrates a demo where up to six different categories of clothing are sequentially changed. Additionally, Fig. 15 and Fig. 16 display additional results of MMTryon.

## A.5 LIMITATION AND FUTURE WORK.

While our method demonstrates strong performance, it still has certain limitations. In the data generation process, our method is influenced by the limitations of pretrained models, making it challenging to produce data that meets the requirements for very fine parts, such as cuffs and collars. This restricts our ability to generate detailed components. Moving forward, we may focus on fine-tuning large models to construct a more freely detailed and fine-grained dataset, aiming to enhance

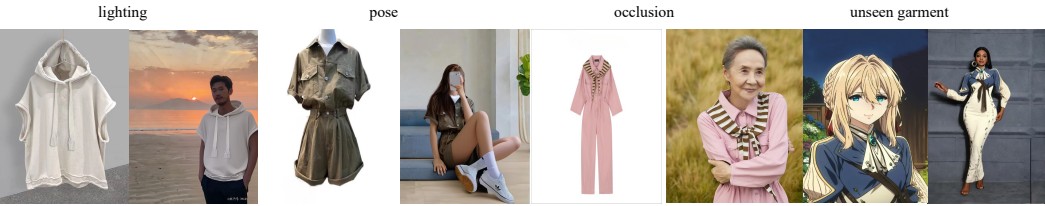

source garment MMTryon* MMTryon    source garment MMTryon^ MMTryon    top bottom MMTryon$ MMTryon

(a) Comparison with MMTryon w\o text query loss    (b) Comparison with MMTryon without the diffusion loss in garment encoder    (c) Comparison with MMTryon without the augmented dataset

Figure 10: Additional ablation study.

lighting    pose    occlusion    unseen garment

Figure 11: Additional results for the case of handling Variations in Input Images.

the upper limit of our model. Additionally, texture consistency remains a challenging issue due to the limitations of the VAE of the Stable-Diffusion-based model. We are committed to addressing this limitation in future work by exploring advanced methods to improve the representation and generation of textures in our model.

## A.6 SOURCE CODE

The complete source code will be made publicly available upon acceptance.

## A.7 SOCIETAL IMPACT.

As most generative approaches, MMTryon can be used for malicious purposes by generating images that infringe upon copyrights and/or privacy. Given these considerations, responsible use of the model is advocated.

## A.8 INFERENCE SETTING

In this section, we provide a detailed explanation of the inference process for our method. In the single-image setting, we use a single garment image, a protection region map, and the corresponding caption as inputs. For the multi-image setting, we use multiple garment images, such as tops, pants, shoes, and bags, along with a protection region map and the corresponding caption. When a reference image for a specific region is not provided, our model generates a suitable garment randomly for that region. This capability ensures that the try-on process remains flexible and produces coherent results even in the absence of complete input references.

Finally, the inference time and GPU RAM requirements are as follows:

- For 1 reference: 13.9s, 11.9GB
- For 2 references: 16.7s, 13.8GB
- For 3 references: 19.0s, 14.2GB
- For 4 references: 22.07s, 14.6GB
- For 5 references: 25.07s, 15.5GB
- For 6 references: 28.02s, 16.0GB

## A.9 MORE RESULT FOR TEXTURE AND COMPARISION

In this section, we present experimental results to demonstrate the effectiveness of our method in handling complex textures. As show in Fig. 17, while our approach may struggle to reproduce

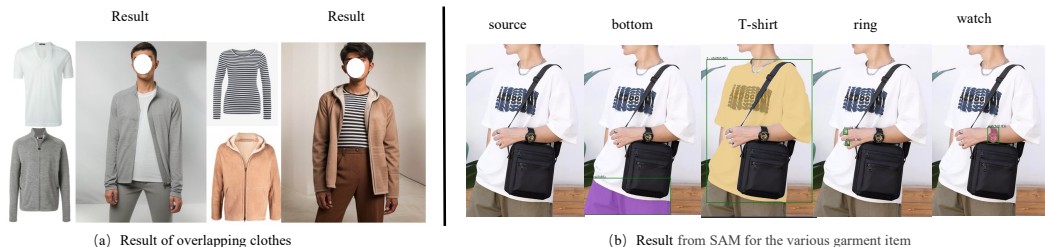

Figure 12: Additional comparision of MMTryon to the baselines .

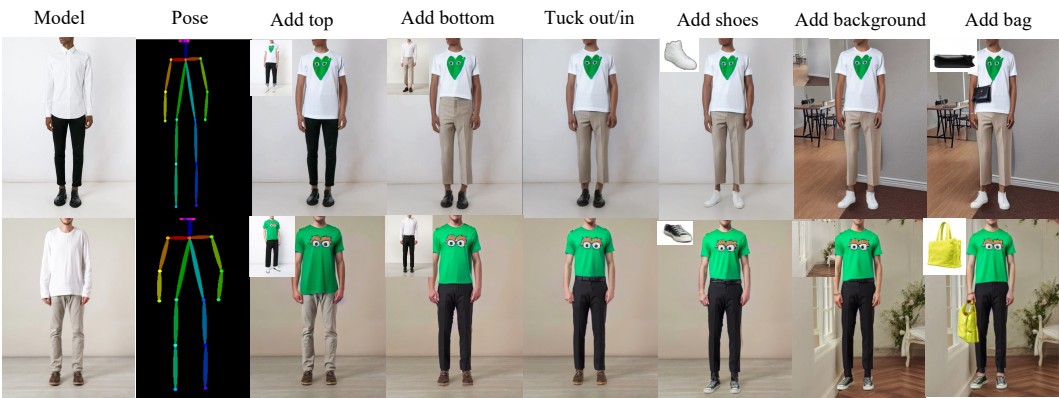

(a) Result of overlapping clothes

(b) **Result** from SAM for the various garment item

Figure 13: More result for result and data process .

Figure 14: More results of MMTryon. Our method demonstrates high quality result in a large variety of scenarios and precise control over dressing styles through instruction-based manipulation.

extremely fine details, such as very small text, it performs well in most scenarios involving intricate textures.

To validate the effectiveness of our method, we compare it with several baseline approaches. As shown in Figure 18, we selected examples with complex textures, including patterns, logos, and detailed clothing designs. The results illustrate that our method successfully restores most texture details, achieving a high level of fidelity in the generated try-on outputs.

For certain baselines where direct comparison is not feasible, we have analyzed their approaches based on their respective papers to highlight the differences: M&M VTON Zhu et al. (2024) demonstrates promising control capabilities; however, based on the results presented in their paper, the supported poses and clothing categories appear to be limited. We look forward to the release of more testable results for further analysis. MV-VTON Wang et al. (2024) is a multi-view try-on method, which focuses on a different research area and is not directly related to our work. Wear-Any-WayChen et al. (2024) introduces an innovative point-based control mechanism for style, but its results are limited to single-image cases. While the control method is noteworthy, it lacks broader evaluation scenarios.

These insights help position our method in the context of existing works and emphasize its contributions to advancing virtual try-on tasks.

| Model | Model | Top | Bottom | Result |
|-------|-------|-----|--------|--------|

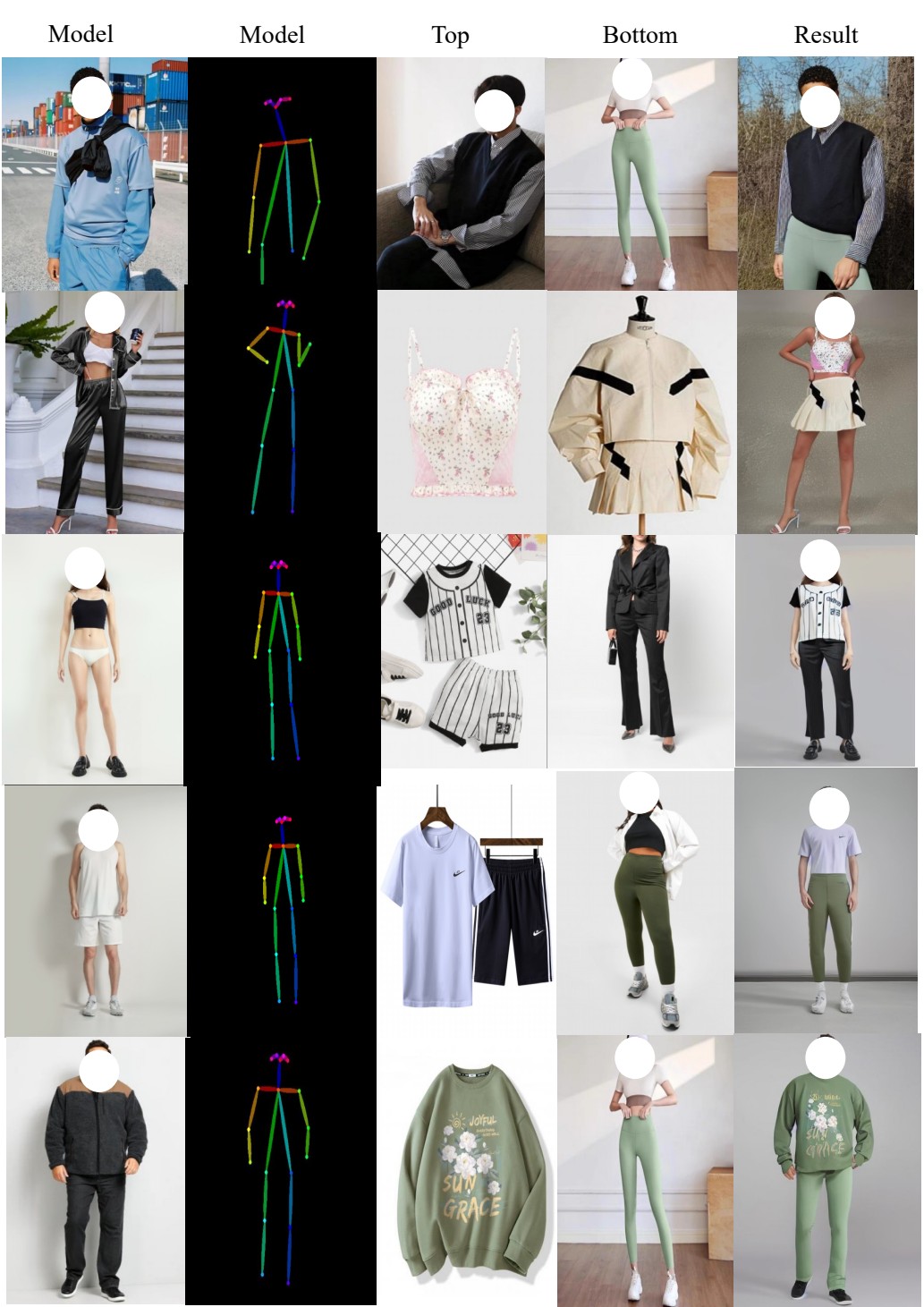

Figure 15: More results of our MMTryon. Please zoom in for more details.

| Garment | Model | Result | Garment | Model | Result | Garment | Model | Result |

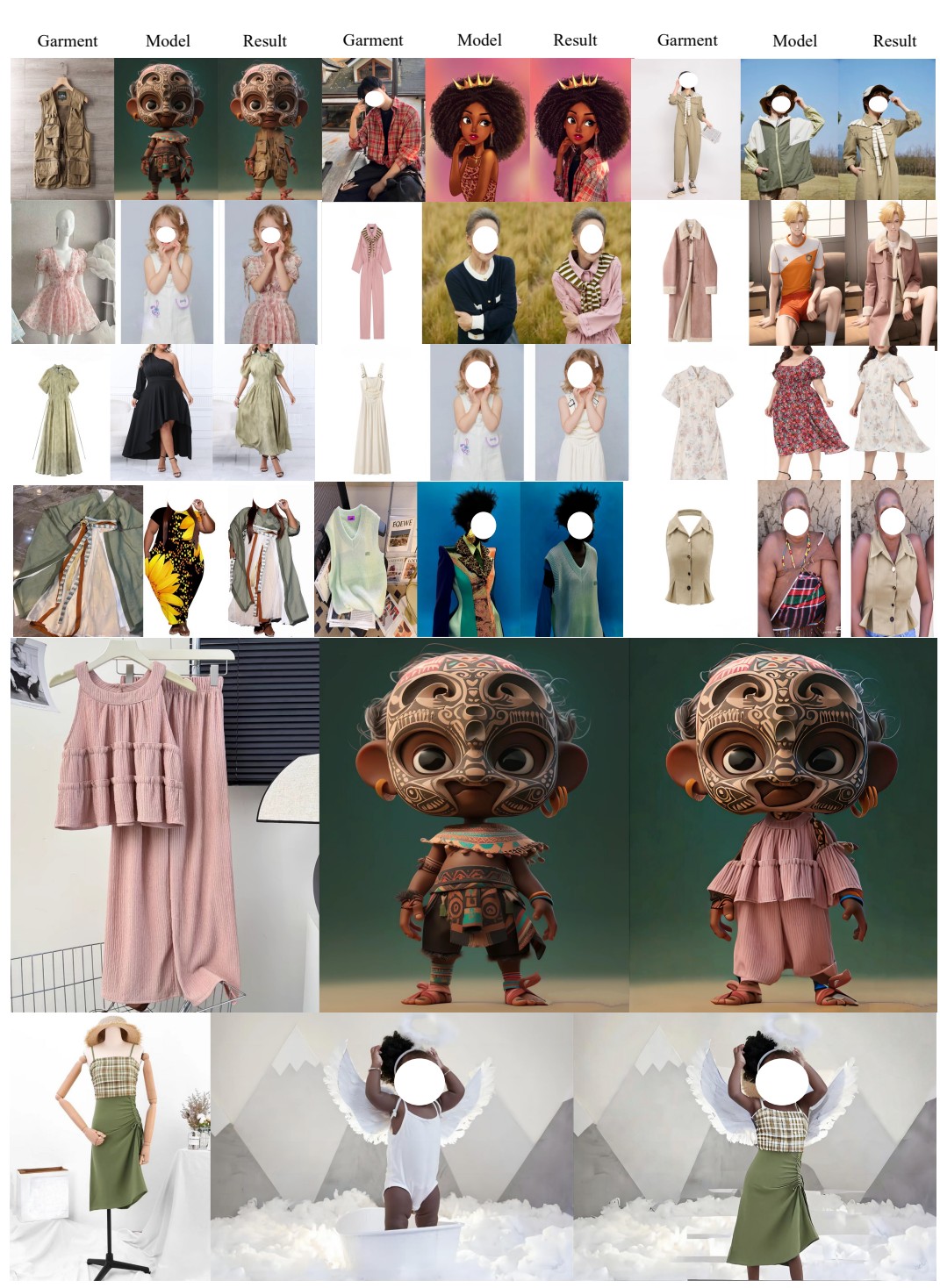

Figure 16: More results of our MMTryon. Please zoom in for more details.

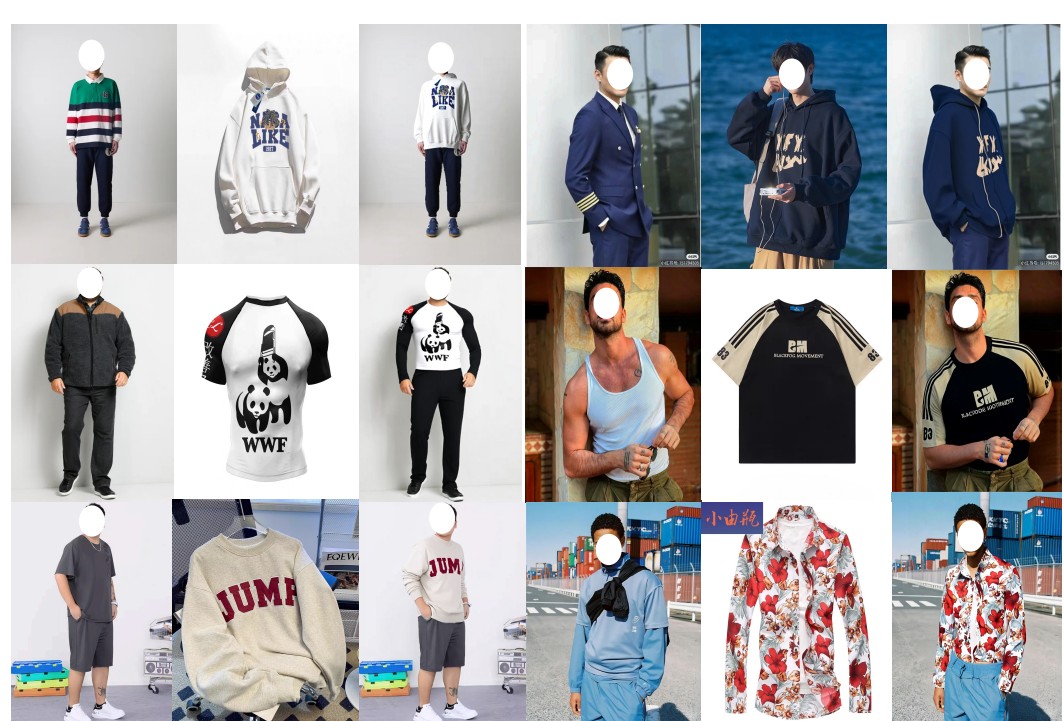

Figure 17: More results of our MMTryon. Please zoom in for more details.

model       garment       Kolors-Virtual-Try-On       MiaoYa       Ours

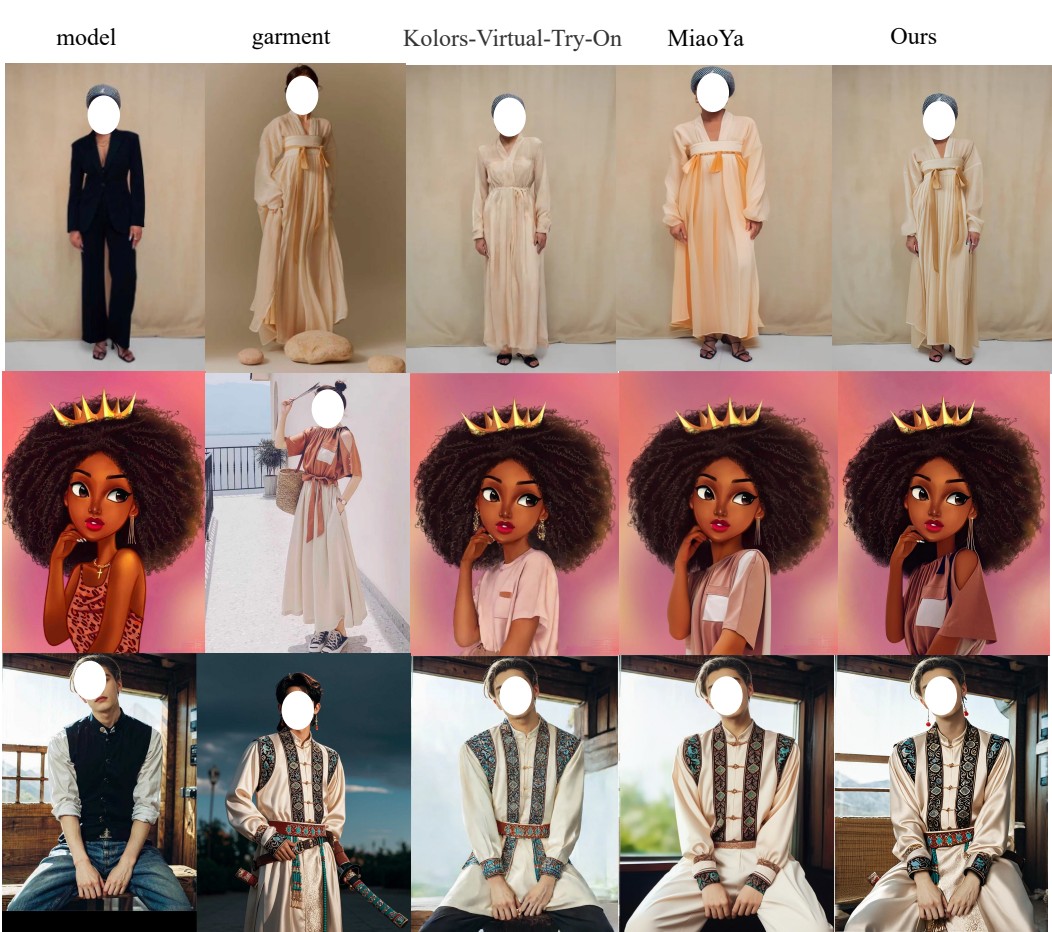

Figure 18: More results of our MMTryon. Please zoom in for more details.

