# OpenReview forum: "MMTryon: Multi-Modal Multi-Reference Control for High-Quality Fashion Generation"
_ICLR.cc/2025/Conference — Submitted to ICLR 2025_

### Official Review · Reviewer_rDXZ · 2024-11-01

**Soundness:** 2
**Presentation:** 2
**Contribution:** 2
**Rating:** 3
**Confidence:** 5

**Summary:**

This paper combines multi-modal implementation of VITON.
Three problems are proposed and solved by the designed data set and multi-model instruction attention.
The experimental results show that the data in the wild well.

**Strengths:**

1. MMTryon is able to combine multiple garments for fitting, while also allowing the fitting effect to be manipulated by text commands.
2. A scalable data generation pipeline and a specially designed clothing encoder are introduced.
3. Eliminates the need for any prior segmentation network during the training and reasoning phases.

**Weaknesses:**

1. The definition of the task is not very clear, and similar work may already exist, leading to unclear motivation. Related work needs to be strengthened.
2. multi-modal multi-reference attention is just a simple extension of reference attention [1], and it is not clear how to decouple and align the details.
3. The quantitative and qualitative results are not complete, especially the texture details are not so good.
4. The data set is the highlight of this paper, but it doesn't seem to claim to be open data, so I don't think it's a contribution either.
---
[1] Hu L. Animate anyone: Consistent and controllable image-to-video synthesis for character animation[C]//Proceedings of the IEEE/CVF Conference on Computer Vision and Pattern Recognition. 2024: 8153-8163.

**Questions:**

1. The paper makes overstated claims, such as line 96 where it states, "is the first model to support the multi-modal compositional try-on task." However, multi-modal compositional try-on is already supported by M&M VTO [1], AnyFit [2], and VITON-DiT [3]. I don't think the relevant work is sufficient. And I recommend the authors explicitly compare these previous works in terms of their multi-modal capabilities to clarify if the proposed approach indeed offers novel advantages or is simply an extension of existing techniques.

2. The paper lacks clear motivation and has weak contributions. For example, Question 1 has already been explored, but the authors do not clearly articulate its relevance or contribution here. Additionally, texture consistency—an essential issue in VTON—seems neglected. I suggest the authors further clarify the connection between their research questions and contributions and add a discussion on texture consistency, as this factor is critical to the realism of virtual try-on. This improvement would enhance the clarity and relevance of the study.

3. The novelty is limited. The garment encoder appears to use cross-attention, which raises the question of whether there is anything unique about it. I recommend the authors explain how their garment encoder differs from or improves upon standard cross-attention mechanisms, especially if there are any specific design or functional enhancements, to demonstrate genuine innovation.

4. Serious concerns arise from fusing text and image features directly without decoupling, which may lead to unintended consequences, yet the paper does not address this. I suggest the authors provide a more thorough justification for this fusion strategy, as bypassing decoupling could compromise the model’s interpretability or adaptability. Discussing the rationale behind this choice and evaluating potential pitfalls would strengthen the explanation.

5. The paper lacks thorough comparison methods. In particular, quantitative results for single-garment cases are not provided for methods like Outfit Anyone, Kolors-Virtual-Try-On [4], Wear-Any-Way [5], and MV-VTON [6]. Additionally, Figure 6 does not display clear advantages. The omission of quantitative comparisons with key methods weakens the study’s validity. I suggest including comparative performance metrics with these established models in Table 1 and providing more insightful analysis of Figure 6 to support the claimed improvements. Besides,  Under multi-modal conditions, Table 7 suggests that background is not controllable. However, Imagedressing [7] has achieved background control, yet it is not referenced or discussed. If background control is indeed relevant, I suggest that the authors provide a direct comparison with Imagedressing or explicitly state if background control is beyond the scope of their study to give readers a clearer understanding of the method’s applicability.

6. Texture details (such as Pants, logo changed) appear poorly rendered across multiple cases, including those in the appendix. I recommend identifying specific areas with poor texture rendering and offering suggestions to improve texture detail quality, as this would enhance the realism and appeal of the model’s outputs.

7. The implementation details concerning SDXL on line 295 and SD 1.5 on line 352 are confusing. I suggest that the authors clarify why they employ different Stable Diffusion versions in various pipeline segments and explain the implications for model outputs to improve method transparency.

8. Adding results under similar conditions, such as challenging cases with letters, numbers, and diverse textures, is advised. The paper would benefit from showcasing the model’s performance on complex garments, such as those with letters or intricate textures, as these are realistic challenges in virtual try-on systems.


Other details:
1. In Table 2,  '0.7' should be '0.70'
2. Formulas 1 and 2 have commas at the end, while formulas 3 and 4 are missing.
---
[1] Zhu L, Li Y, Liu N, et al. M&M VTO: Multi-Garment Virtual Try-On and Editing[C]//Proceedings of the IEEE/CVF Conference on Computer Vision and Pattern Recognition. 2024: 1346-1356.

[2] Li Y, Zhou H, Shang W, et al. AnyFit: Controllable Virtual Try-on for Any Combination of Attire Across Any Scenario[J]. arXiv preprint arXiv:2405.18172, 2024.

[3] Zheng J, Zhao F, Xu Y, et al. VITON-DiT: Learning In-the-Wild Video Try-On from Human Dance Videos via Diffusion Transformers[J]. arXiv preprint arXiv:2405.18326, 2024.

[4] https://huggingface.co/spaces/Kwai-Kolors/Kolors-Virtual-Try-On.

[5] Chen M, Chen X, Zhai Z, et al. Wear-any-way: Manipulable virtual try-on via sparse correspondence alignment[J]. arXiv preprint arXiv:2403.12965, 2024.

[6] Wang H, Zhang Z, Di D, et al. MV-VTON: Multi-View Virtual Try-On with Diffusion Models[J]. arXiv preprint arXiv:2404.17364, 2024.

[7] Shen F, Jiang X, He X, et al. Imagdressing-v1: Customizable virtual dressing[J]. arXiv preprint arXiv:2407.12705, 2024.

---

> ### Author Response · Authors · 2024-11-21
>
> # (Part1)
> We sincerely thank you for your insightful and constructive feedback. Below, we address each of your comments in detail, grouped into relevant subsections.
>
> # Overstated Claims and Comparison with Prior Work (Q1)
>
> We appreciate your concern regarding the claims made in our paper. However, we would like to clarify that we have not overstated our contributions. The multi-modal compositional try-on task we define accepts multiple garments and text inputs, which was not addressed in prior works at the time of our paper’s submission.
>
> Regarding the examples you provided:
>
> - **M&M VTON:**
>   While it accepts text and garments, it does not support free-form language input. For instance, it cannot handle layout editing tasks as required for style specification and also not user-defined backgrounds. However, we have demonstrated this capability in the appendix, where MMTryon effectively incorporates instructions (see Figure 14 in Appendix).
>
> - **AnyFit:**
>   This work was published after our submission. Furthermore, it does not support controllable wearing styles, and its compositional garment capabilities are limited to flat-lay clothing pairs, such as tops and bottoms, without further interaction.
>
> - **VITON-DiT:**
>   This is primarily a video try-on model and does not support text-based inputs. As such, we believe it is not directly comparable to our approach.
>
> We hope this explanation clarifies the distinctions between our method and the cited works. We have also added this discussion to strengthen our related work section at Line 139. Thank you for considering our response.
>
> # Motivation, Contributions, and Texture Consistency (Q2, W1, W2, W3)
>
> We appreciate your concerns regarding the contributions of our work. Here, we would like to restate and clarify our key contributions:
>
> - **Multi-modal compositional try-on task:**
>   Our work introduces the first model to support this task. MMTryon enables the combination of one or multiple garments for try-on while allowing manipulation of try-on effects through textual commands.
>
> - **Scalable data generation pipeline and garment encoder:**
>   We propose a scalable data generation pipeline and a specially designed garment encoder, which eliminates the need for prior segmentation networks during both training and inference. Instead, our approach relies solely on textual inputs, images of individual garments, or model images to identify the areas of interest.
>
> - **Extensive benchmarks and state-of-the-art performance:**
>   We conduct comprehensive experiments on public try-on benchmarks and in-the-wild test sets, achieving state-of-the-art results.
>
> Regarding the first contribution, we have already addressed the comparison with related works in our response to Q1. Additionally, our method does not require parsing as input, whereas existing works rely heavily on it. To the best of our knowledge, no current approach has eliminated this dependency.
>
> Further, while we do not explicitly position texture handling as one of our primary contributions, our robust garment encoder provides strong texture representations.
> To illustrate this, we have added a detailed discussion on texture consistency supported by extensive results related to texture quality in the appendix (see Section A.9), where we compare our approach to the open source baselines.
>
>
> # Novelty of the Garment Encoder (Q3)
>
> We appreciate your concerns regarding the novelty of our garment encoder. Here, we would like to clarify our contributions and address the points raised:
>
> Our garment encoder introduces a novel **text query loss**, which enhances garment representation during training. This loss formulation eliminates the influence of parsing networks, enabling our method to be *parsing-free*.
> Additionally, we leverage data augmentation to further improve the garment encoder's ability to represent diverse clothing items effectively.
> While we do employ cross-attention mechanisms, we acknowledge that we have not introduced specific innovations in their design. Instead, our contributions focus on achieving a *parsing-free* capability, which we believe is a significant advancement and distinguishes our work from existing approaches.
>
> To the best of our knowledge, no prior work has addressed the topic of parsing-free virtual try-on in this manner. Furthermore, we have conducted extensive ablation studies to validate the effectiveness of our parsing-free design and text query loss. We kindly request you to review the section as it provides a deeper understanding of our contributions.

---

> > ### Author Response · Authors · 2024-11-21
> >
> > # Part 2
> > # Fusion of Text and Image Features (Q4)
> >
> > We appreciate your attention to this aspect of our work. We would like to point out that the feature injection strategy we employed is commonly used in current multi-modal large models.
> > Specifically, the image embeddings in our approach pass through a learnable mapper layer before encoding. This ensures proper alignment between the image and text features, addressing potential discrepancies and improving feature integration.We believe this design effectively balances simplicity and performance while maintaining coherence between modalities. Thank you for highlighting this point.
> >
> > # Comparison Methods and Missing Metrics (Q5)
> >
> > We appreciate your concerns regarding the comparisons conducted in our work. Following your advice, we have added additional comparisons to Wear-Any-Way and AnyFit (see Table 1). Additionally, we have provided a qualitative comparison to Kolors-Virtual-Try-On in the Appendix. Quantitative comparisons to Kolors-Virtual-Try-On were unfortunately not possible due to usage restrictions of their provided API.
> >
> > Regarding the comparison with OutfitAnyone in Figure 6, we would like to highlight the following observations based on Figure 6:
> > In the first, third, and fourth images, our method demonstrates superior texture consistency compared to the baseline, accurately preserving fine-grained details in the garments.
> > In the second image, OutfitAnyone incorrectly restores the mannequin's white skin, which reflects the limitations of parsing-based methods. In contrast, our approach correctly restores the skin, showcasing the robustness of our model in such scenarios.
> > Please note that the other baselines listed do not provide open-source code, making direct comparisons infeasible. For these cases, we instead provide a discussion of the differences and their shortcomings (see the updated Related Work section and Appendix A.9).
> >
> > While modifying the background is certainly of interest for some tasks, we did not consider it as the main objective of our study, where the focus is on the try-on generation and have therefore not included background instructions in our main experiments. Note, however, that MMTryon is able to change the background if provided with the corresponding instruction (see Figure 14 in the Appendix).
> >
> > # Texture Rendering and Complex Cases (Q6, Q8)
> >
> > We appreciate your observations regarding the texture inconsistencies in certain results. The cases where textures appear poorly rendered are primarily full-body pictures where the text occupies a small proportion. For such very small text, the generation quality is impacted by the compression issue inherent to the Stable Diffusion VAE.
> > However, we would like to emphasize that our method has demonstrated strong performance in challenging scenarios. We have provided more results related to texture quality (see A.9).
> >
> > This limitation is also common across Stable Diffusion-based methods. We propose several approaches to mitigate this problem, such as fine-tuning the VAE and increasing the resolution, and we have included this point in our limitations and future work section to ensure it is systematically addressed in subsequent studies.
> >
> > # Implementation of SDXL and SD 1.5 (Q7)
> >
> > We appreciate your inquiry regarding the use of SDXL and SD 1.5 in different parts of our pipeline. Here is the rationale behind this design choice:
> >
> > - **SDXL for Data Generation:**
> >   SDXL typically produces higher-quality images, making it highly suitable for dataset construction. By leveraging SDXL, we ensure that our dataset is of superior quality, which contributes to the robustness of our method.
> >
> > - **SD 1.5 for Model Training:**
> >   While SDXL provides high-quality images, running multi-reference tasks on SDXL requires substantial GPU resources, which exceeds our computational capabilities. As a result, we use SD 1.5 as the base model for training. Despite this, our experimental results demonstrate that our method remains highly effective and achieves strong performance.
> >
> > We hope this explanation clarifies the reasoning behind our choices and demonstrates the effectiveness of our approach under the given resource constraints. Thank you for raising this important point.
> >
> > # Ethical Concerns
> >
> > We acknowledge your concerns regarding legal compliance and would like to emphasize that the proprietary dataset was only collected from e-commerce websites that we have explicit permissions to access (see Appendix A1). Further, we acknowledge that all generative models have the potential to be used for malicious purposes (see Appendix A7); however, this approach is not more harmful than alternative generative models.
> >
> > # Conclusion
> >
> > We are grateful for your valuable feedback, which has helped us identify areas for improvement and clarification and allowed us to address them in the revised manuscript.We hope this structured response together with the revised manuscript adequately addresses your concerns and suggestions.

---

> ### Comment · Reviewer_rDXZ · 2024-11-22
> **Hope the author can explain further**
>
> Thank you for the detailed response.
>
> （1）*Overstated Claims*
> - **Please provide further clarification on this point.** M&M VTON  supports layout editing tasks, as illustrated in Figure 5 of the paper.
>
> - **I do not fully agree with the statement.**  AnyFit： "This work was published after our submission." While it may be considered concurrent work, it cannot be denied that the official publication date is May 2024.
>
> -  **I find the statement "which eliminates the need for prior segmentation networks during both training and inference" unclear.** Since SAM was used during the training phase, this seems contradictory. Furthermore, CatVTON [1] also emphasizes being a mask-free approach,
>
> （2）*Incremental improvement*
>
> It appears that incorporating SAM during training for segmentation and applying a text query loss for supervision resembles a cascade network structure. This seems more like an incremental improvement rather than a novel framework. **Based on this, the main contribution of the work seems to lie in the text query loss rather than the Garment Encoder.**
>
> **To summarize my observations:**
> - The first contribution feels somewhat overstated.
> - The second contribution comes across as incremental.
> - Simply achieving SOTA performance does not qualify as a significant contribution on its own. Moreover, in A.9, the results on complex textures, such as text or letters, are still quite rough and lack refinement.
>
> I would appreciate further clarification and discussion from the authors on these points.
>
> [1] Chong Z, Dong X, Li H, et al. CatVTON: Concatenation Is All You Need for Virtual Try-On with Diffusion Models[J]. arXiv preprint arXiv:2407.15886, 2024.

---

> > ### Author Response · Authors · 2024-11-23
> >
> > Thank you for your feedback. We appreciate the opportunity to discuss and clarify some of the issues you raised. Below is our detailed response:
> >
> > # Overstated Claims
> > **1. Regarding M&M VTON**:
> >
> > We acknowledge that Figure 5 in the M&M VTON paper demonstrates some basic layout editing tasks. However, the approach struggles with more advanced editing tasks, as discussed in the limitations section of their paper (see the discussion in the last section on page 7 of the original paper). Our approach instead is able to perform these editing tasks, as demonstrated in Fig. 13 (a). We will revise our wording to make this distinction clearer. Note additionally, that M&M VTON similar to other approaches is restricted through their use of a prior expert segmentation model that is specifically trained for the try-on task, which restricts its versatility. This will be discussed in more detail in the following sections of this response.
> >
> >
> > **2. Regarding AnyFit**:
> >
> > We agree that it will be better to refer to AnyFit as concurrent work as the ArXiv preprint was released in the end of May 2024. We will amend the statement. However, please note that the main point still holds, that AnyFit's compositional garment capabilities are limited to flat-lay clothing pairs, such as tops and bottoms, without further interaction.
> >
> > **3. Regarding the statement about the prior segmentation network**:
> > We would like to clarify that the prior segmentation networks refer to expert models that are specifically trained for try-on tasks, as what is done in current try-on approaches. This model typically segments the human body into 18 categories, which often introduces fixed annotation errors that subsequently affect generation quality.
> >
> > In contrast, we adopted SAM, a large-scale model, for this work. In the next section, addressing your second point, we will elaborate on the contribution of our paper and how our design choices were motivated.
> >
> > # Incremental improvement:
> > We believe it is unreasonable to constrain try-on models with fixed-category parsing. This limits the flexibility of try-on tasks and restricts the model’s development toward more open-ended and versatile applications. Our goal was to enable our method to comprehend concepts such as “top” and “bottom.”
> >
> > During the training of the garment encoder, we leverage large-scale models to provide such priors. However, this alone does not fully achieve parsing-free capability. To address this, we implemented a second-stage data construction and training process. This involves showing the model variations of a person wearing different tops but the same bottom.
> >
> > Our parsing-free capability, therefore, relies on supervision from large models and learning contextual reasoning through diverse garment compositions. This approach aligns with the core philosophy behind the success of modern large language models.
> >
> > Consequently, we emphasize that our second contribution is not the garment encoder alone but rather the overall design of “a scalable data generation pipeline and a specially designed garment encoder.” We hope this explanation clarifies our intent and the rationale behind our design. While this is an initial exploration, it reflects our thinking and efforts toward enabling more flexible try-on models in the future.
> >
> > Additionally, our parsing-free approach differs fundamentally from CATVTON. While CATVTON adopts a mixed mask-based and mask-free training approach, it lacks the mechanisms to compel the model to reason about try-on context effectively. Their approach is instead more related to the conversion of garment-to-person to person-to-person data, which we briefly describe in appendix A1 and in our case is a pre-processing step to generate person-to-person data.
> >
> > # Summary of Response:
> > - **First contribution**: We have provided a response to clarify our first contribution and hope you find it satisfactory.
> > - **Second contribution**: We have explained in detail why our contributions are designed this way, emphasizing our focus on future-proofing try-on models.
> > - **Texture limitations**: We acknowledge your observation regarding texture issues and have already noted this limitation in our paper. We agree that texture optimization requires further research, and we acknowledge that no existing work has completely resolved this challenge. Our primary contribution, however, lies not in meticulous texture exploration but in pushing the boundaries of try-on models toward a more flexible, free-prompt-driven paradigm, demonstrating the potential under this novel setting. Finally, our focus on enabling more versatile, open-ended try-on tasks addresses limitations of prior methods and highlights the forward-looking nature of our contributions.
> >
> > We hope this response addresses your concerns comprehensively and clarifies the motivations and contributions of our work. Please feel free to provide further feedback or suggestions.

---

> > > ### Author Response · Authors · 2024-11-27
> > >
> > > Dear Reviewer rDXZ
> > >
> > > Thank you for your thoughtful review of our paper and for providing valuable comments to enhance its quality. As the deadline for updating the PDF draws near, we look forward to your feedback on our response. If you have any additional comments, we would be glad to offer further clarifications. Thank you once again for your time and efforts!

---

> ### Author Response · Authors · 2024-12-01
>
> Dear Reviewer rDXZ,
>
> As the discussion phase is ending, we would like to thank you again for your feedback, which has allowed us to improve our work. We would like to briefly summarize the main changes and clarifications that we have provided that addressed the concerns of Reviewer mfjz and qJg5 and hopefully have addressed your concerns as well:
>
> - Clarified our contributions and their novelty in the context of related work (comments above and Part 2)
> - Revised the related work section to include additional related work on controllable generation for virtual try-on (Part 2)
> - Improved the description of the inference phase (Appendix A.8)
> - Added additional details about the dataset (Appendix A.1)
> - Provided additional results and discussion on texture consistency (Appendix A.5 and A.9)
> - Provided additional comparisons to related work (Table 1 and Appendix A.9)
>
> We hope that these modifications and clarifications have addressed your concerns. If there are any remaining issues or points, we would be more than happy to provide further responses.
>
> Best regards,
>
> The Authors

---

### Official Review · Reviewer_qJg5 · 2024-11-01

**Soundness:** 4
**Presentation:** 4
**Contribution:** 4
**Rating:** 10
**Confidence:** 5

**Summary:**

This paper introduces MMTryon, a multi-modal, multi-reference Virtual Try-On (VITON) framework capable of generating high-quality, compositional try-on results from text instructions and multiple garment images. MMTryon addresses three main limitations in existing methods: supporting multiple garments, specifying dressing styles, and reducing dependency on segmentation models. It achieves this through a novel multi-modality attention mechanism that combines garment details from images with style cues from text, and by using a parsing-free garment encoder alongside a scalable data pipeline to eliminate the need for segmentation. Extensive experiments demonstrate MMTryon’s superior performance in multi-item and style-controllable try-on tasks, offering new potential for applications in the fashion field.

**Strengths:**

1. The paper introduces a unique attention mechanism that enables multi-item try-on and customizable styling, addressing limitations in previous models.
2. The paper adopts a novel approach by eliminating segmentation dependency using a parsing-free garment encoder and scalable data pipeline. This method presents a practical solution to common issues in virtual try-on tasks, improving model efficiency and minimizing artifacts.
3. The paper is well-organized, and there are a lot of ablation study in main paper and appendix to demonstrate impact of each component in the proposed approach.

**Weaknesses:**

1. The Related Work section lacks recent studies on control and try-on methods,
2. More detail information about dataset should be provided.
3. in some case, the fine texture on the garment still cannot be recovered completely,

**Questions:**

1. Could you provide more information about the e-commerce dataset used in the experiments?
2. Do you have any plans to open-source the dataset and the data processing pipeline? Making these resources publicly available could greatly benefit research in this area and facilitate further advancements in virtual try-on methods.

---

> ### Author Response · Authors · 2024-11-21
>
> We sincerely thank you for the review, which has allowed us to further strengthen our work, and we are glad that you appreciate the contributions of our work.
> # Related Work Revisions (W1)
> Thank you for pointing out the concern regarding the lack of related work. We have revised the Related Work section in the main text to include discussions of the latest controllable generation methods for virtual try-on, such as M&M VTON and others (see Line 139).
>
> # More Detail of Our E-Commerce Dataset (Q1)
> Thank you for pointing out that some of the dataset details are not provided in sufficient detail. We have provided a description of the dataset collection process in Appendix A1 and will release the dataset construction code to ensure full transparency on how the dataset is constructed from the image dataset.
>
> Additionally, in the revised version of Appendix A1, we have provided more details on the types of tagging models and prompts that were used to improve the dataset's diversity and quality.
>
> # The Plan for Code and Dataset (Q2)
> The complete dataset collection and construction process, as well as the code and model weights, will be released upon acceptance.
>
> # Fine-detaile discovery(W3)
> We appreciate your attention to the issue of texture consistency, which indeed remains a challenging problem. The cases where textures appear poorly rendered are primarily full-body pictures where the text occupies a small proportion. For such very small text, the generation quality is impacted by the compression issue inherent to the Stable Diffusion VAE.
>
> However, as shown in the additional texture results provided in the appendix (see the new Section A.9), our method demonstrates strong performance in most cases. While we acknowledge this as a limitation in specific scenarios, we will consider this an important direction for our future work to further enhance texture consistency. We have updated the limitation section (A.5) accordingly.
>
> Thank you for your thoughtful feedback.

---

> > ### Comment · Reviewer_qJg5 · 2024-11-22
> >
> > Thanks for the author's reply. The author's reply answered my doubts. I will change my rating to 10.

---

> > > ### Author Response · Authors · 2024-11-22
> > >
> > > Thank you again for your positive view of our work and for your suggestions and comments, which have allowed us to improve the manuscript. We are glad that we were able to address your doubts in our rebuttal.

---

### Official Review · Reviewer_mfjz · 2024-11-02

**Soundness:** 4
**Presentation:** 4
**Contribution:** 4
**Rating:** 8
**Confidence:** 5

**Summary:**

The paper introduces MMTryon, a model that integrates a multi-modality and multi-reference attention mechanism to enhance virtual try-on tasks by combining garment details from reference images with dressing styles from text guidance. The model is designed for multi-garment try-ons and does not require segmentation networks, thanks to a new parser-free garment encoder and a scalable data generation pipeline. MMTryon achieves superior performance in single and multi-garment try-on tasks, addressing limitations in previous models such as single-garment constraints, fixed dressing styles, and segmentation dependencies.

**Strengths:**

1. The paper introduces innovative multi-modal and multi-reference attention mechanisms, allowing for multi-garment try-ons with customized, fine-grained text guidance.

 2. Leveraging a pretrained garment encoder, MMTryon can capture detailed features of garments specified by text, improving model precision and contributing to advancements in the field.

 3. Extensive qualitative and quantitative evaluations demonstrate MMTryon’s superior performance compared to state-of-the-art methods, with high-quality visual results showcasing realistic try-on effects and robust editing capabilities.

 4. The proposed data generation pipeline creates a comprehensive paired multi-garment dataset without segmentation, broadening the scope of try-on tasks and supporting complex garment compositions.The parsing-free approach in this article presents a promising direction, contributing to the advancement of try-on technology toward more practical applications.

 5.This article is well-written

**Weaknesses:**

1.some descriptions could be clearer to avoid reader confusion. For example, in Fig. 3, labeling the UNet as “MMTryon UNet” rather than specifying it as a pretrained SD UNet might mislead readers at this training stage. Similarly, terms like “w/o TQL” and “w/o MRA” in Fig. 8 are somewhat unclear;

2.the inference process requires more detailed explanation, especially given the model’s support for multiple tasks. Currently, some detail of inference lack explaination, which may lead to unintended changes in specific garments, such as unexpected alterations in lower garments shown in the first image.

**Questions:**

1.As mentioned in the weaknesses, please explain why the pants in the last rows of the first image changed unexpectedly.

2.the method is good，Are there any plans to release the code and model to support the  research community?

3.What is the inference time and required GPU memory for MMTryon in different task?

---

> ### Author Response · Authors · 2024-11-21
>
> Thank you for your detailed and thoughtful feedback on our paper. We are pleased that you recognize the strengths of our work. We will address your concerns, including weaknesses and questions (abbreviated as W and Q), point by point:
>
> # Single Garment Try-on Control (Q1,W2)
> In the two images displayed in Figure 1, the issue arises because, during testing, we did not specify the lower garments, resulting in their free generation. When we provide both the original model image and the prompt for the lower garment, we are able to control the lower garments effectively. In the revision of our paper, we have changed it and we comment on this behavior in the new section on Inference (Appendix A.8).
>
> # The Plan for Code and Dataset (Q2)
> We will open-source all the code and model weights to contribute to further research in this area.
>
> # Inference Time and GPU RAM (Q3)
> The inference time and GPU RAM requirements for our model are as follows:
>
> - For 1 reference: **13.9s**, **11.9GB**
> - For 2 references: **16.7s**, **13.8GB**
> - For 3 references: **19.0s**, **14.2GB**
> - For 4 references: **22.07s**, **14.6GB**
> - For 5 references: **25.07s**, **15.5GB**
> - For 6 references: **28.02s**, **16.0GB**
>
> We have revised the paper and added this information to Appendix A.8.
>
> # Figure Descriptions(W1)
> We appreciate your attention to detail. In the revised version, we have improved these descriptions.
>
> Thank you again for your valuable feedback, which has helped us further improve the clarity and robustness of our work.

---

> ### Comment · Reviewer_mfjz · 2024-11-24
> **Comments by the reviewer**
>
> Thank you for your detailed response. Your explanation has adequately addressed my concerns. Regarding texture consistency, I also reviewed the comments from the other reviewers, and I agree that texture consistency is indeed a challenging issue. This limitation is partly influenced by resource constraints and the quality of the base model, and I acknowledge that it does not conflict with the contributions emphasized in your paper.
>
> I find the most impressive aspect of your work to be the parsing-free approach. This innovation offers valuable insights for the virtual try-on research community and aligns well with the current trends in AGI. Given these strengths, I will maintain my score.

---

> > ### Author Response · Authors · 2024-11-25
> >
> > Thank you very much for your positive feedback and recognition of our work.  We sincerely appreciate your thoughtful review .Your constructive feedback has been invaluable in helping us refine and improve our work.

---

### Meta-Review · Area_Chair_3UN1 · 2024-12-12

**Metareview:**

The paper receives mixed scores from the reviewers. While the reviewers appreciate the innovative design and extensive experiments, they also raised some concerns including claims overstated, the contributions incremental, and the texture quality limited.

In particular, M&M VTO already demonstrated several of the paper's claims (e.g. multi-modal compositional try-on) and indeed significantly weakens the paper's contributions. While M&M VTO admits it is not designed for this task, it is hard to deny the work can achieve such capability. The authors are encouraged to conduct more complete literature survey prior to the submission.

Moreover, the texture quality (e.g. text) is indeed limited in many cases (e.g. in Fig. 15). It appears that examples shown in M&M VTO (e.g. Fig. 4 in their paper) has better text quality. For qualitative comparisons to existing methods (Fig. 5 and 6), it seems a bit unclear if the proposed model is more correct than other methods.

Based on the above observations, the authors are encouraged to perform more detailed analysis and comparisons to existing works in the revised version.

**Additional Comments On Reviewer Discussion:**

During discussion, reviewer rDXZ raised several concerns, including overstated claims, incremental contributions, and limited result quality. The authors' rebuttal partially resolves the concerns, but some still remain to be resolved (e.g. claims and result quality), as detailed in the meta-review.

---

### Decision · Program_Chairs · 2025-01-22

Reject